# Theoretical Study and Application of the Reinforcement of Prestressed Concrete Cylinder Pipes with External Prestressed Steel Strands

**Lijun Zhao [1,2], Tiesheng Dou [1,2,\*], Bingqing Cheng [1,2], Shifa Xia [1,2], Jinxin Yang [3], Qi Zhang [3], Meng Li [1,2] and Xiulin Li [1]**

[1] Division of Materials, China Institute of Water Resources and Hydropower Research (IWHR), Beijing 100038, China; jun15297669998@gmail.com (L.Z.); chengbq1994@gmail.com (B.C.); xiasf@iwhr.com (S.X.); limeng@iwhr.com (M.L.); lixl@iwhr.com (X.L.)

[2] State Key Laboratory of Simulation and Regulation of Water Cycle in River Basin, China Institute of Water Resources and Hydropower Research (IWHR), Beijing 100038, China

[3] Beijing Institute of Water, Beijing 100044, China; yangjinxin@biwmail.com (J.Y.); zhangqi@biwmail.com (Q.Z.)

\* Correspondence: doutsh@iwhr.com; Tel.: +86-135-0120-7217

**Abstract:** Prestressed concrete cylinder pipes (PCCPs) can suffer from prestress loss caused by wire-breakage, leading to a reduction in load-carrying capacity or a rupture accident. Reinforcement of PCCPs with external prestressed steel strands is an effective way to enhance a deteriorating pipe's ability to withstand the design load. One of the principal advantages of this reinforcement is that there is no need to drain the pipeline. A theoretical derivation is performed, and this tentative design method could be used to determine the area of prestressed steel strands and the corresponding center spacing in terms of prestress loss. The prestress losses of strands are refined and the normal stress between the strands and the pipe wall are assumed to be distributed as a trigonometric function instead of uniformly. This derivation configures the prestress of steel strands to meet the requirements of ultimate limit states, serviceability limit states, and quasi-permanent limit states, considering the tensile strength of the concrete core and the mortar coating, respectively. This theory was applied to the reinforcement design of a PCCP with broken wires (with a diameter of 2000 mm), and a prototype test is carried out to verify the effect of the reinforcement. The load-carrying capacity of the deteriorating PCCPs after reinforcement reached that of the original design level. The research presented in this paper could provide technical recommendations for the application of the reinforcement of PCCPs with external prestressed steel strands.

**Keywords:** prestressed concrete cylinder pipe; external prestressed steel strands; theoretical study; wire-breakage

## 1. Introduction

A prestressed concrete cylinder pipe (PCCP) contains four components, namely, (1) a concrete core, (2) a steel cylinder lined with concrete (LCP) or encased in concrete (ECP), (3) high strength prestressing wires to withstand the internal high water pressure and external load, and (4) a mortar coating to protect the wires and cylinder against corrosion. The promise of the PCCP lies in its high bearing capacity, strong permeability resistance, and cost-effectiveness. Efficiencies in construction and reductions in fabrication costs have led to the extensive use of PCCPs in the USA, Canada, and China, and have also led to the pivotal development of this pipe. However, these pipes may suffer

from prestress loss caused by wire-breakage. Wire-breakage or rupturing can result in significant losses to society, making the reinforcement of deteriorating pipes essential.

Reinforcement with external prestressed steel strands is regarded as an efficient way of strengthening bridges and beams that are deteriorating due to increased overloading and progressive structural aging [1–5]. Miyamoto A. [6,7] demonstrated the feasibility of applying this prestressing technique to the strengthening of existing steel bridges. Chen S. [8] proposed a finite element model to investigate the inelastic buckling of continuous composite beams that were prestressed with external tendons. Lou T. [9] also concluded that external prestressing significantly improved the short-term behavior of a composite beam. Tan K. H. [2], Aparicio A. C. [10], Park S. [11], and others have presented a series of prototype tests regarding externally prestressed concrete beams and have verified that external tendons can be used to effectively influence beam behavior.

The reinforcement of a PCCP with external prestressed steel strands involves repairing critical pipes with additional external post-tensioning to increase the longevity of problematic PCCP pipelines. The strands are wrapped outside the pipe with a fixed spacing between each strand, according to the service water pressure [12] (Figure 1). A well-known large-scale application of external prestressed strands is in the Great Man-Made River pipelines in Libya [12]. Most of the pipes in this project have an internal diameter of 4.20 m. Authorities have determined that repair of the critical pipes should proceed, with additional external post-tensioning in areas where pipes had burst. The reinforcement of the external prestressed strands on PCCPs has proven to be effective here. This approach is advantageous due to its ability for construction to proceed with no need to drain the pipeline. However, few theoretical studies have been carried out regarding the prestress losses and the mechanism applying external prestressed strands to strengthen PCCPs.

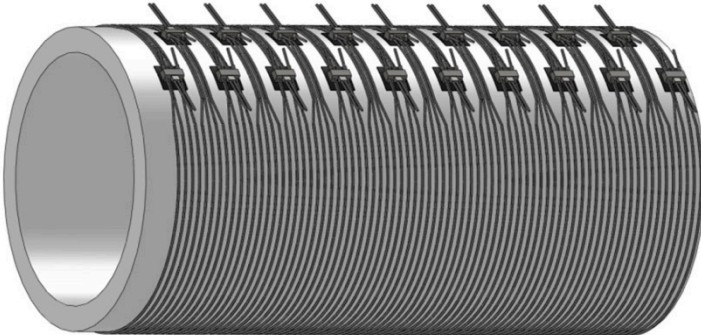

**Figure 1.** Structural drawing of a prestressed concrete cylinder pipe (PCCP) strengthened with prestressed steel strands.

This study introduces a theoretical derivation and investigates the prestress loss of steel strands applied to PCCPs. The normal stress between the strands and the pipe wall is assumed to be distributed as a trigonometric function, instead of uniformly, to estimate prestress losses. The area of the steel strands is determined to meet the requirements of ultimate limit states, serviceability limit states, and quasi-permanent limit states, considering the tensile strength of the concrete core and the mortar coating, respectively. An example calculation of this theory and a prototype test is calculated on the same PCCP to verify the feasibility of this theory. The load response of the pipe before and after the reinforcement process is analyzed.

## 2. Theoretical Derivations

### 2.1. Calculation of Prestress Loss, $\sigma_{st,l}$

The prestress loss persisted during and after the tensioning operation, and can be divided into two categories, namely, instantaneous loss and long-term loss [13–15]. Instantaneous loss, i.e., short-term loss during the tensioning operation, described the prestress losses caused by friction resistance

between the surface of the pipe wall and the steel strands, the anchor deformation, the concrete elastic compression induced by stepwise tensioning operation, and cracks closures. Long-term prestress losses included prestress losses [13,16], while taking into account the materials aging, including the effects of shrinkage and the creep losses of concrete, and the long-term relaxation losses of prestressed steel strands. Types of prestress loss of steel strands applied to PCCPs are illustrated in Figure 2. Since the reinforcement of PCCPs with external prestressed steel strands is a post-tensioning method, the impact of temperature can be removed from consideration when considering the reinforcement of PCCPs with external prestressed steel strands.

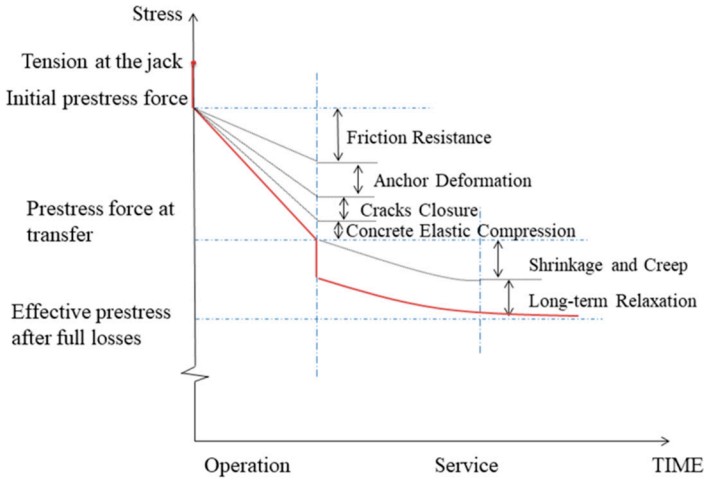

**Figure 2.** Types of prestress loss of steel strands applied to PCCPs.

### 2.1.1. Calculation of Retraction Length, $l_{re}$, and Its Corresponding Retraction Angle, $\theta_{re}$

As far as we know, the stress distribution along the strand is nonlinear. The anchor influenced the prestressed steel strand within a certain length range due to the static friction caused by the retraction of the strand. This length is called the retraction length, $l_{re}$. Strands within the retraction length showed a displacement opposite to the tension direction, which decreases the prestress. The movement trend is demarcated at point $C$, and the stress is redistributed from $ACB$ to $A'CB$ (Figure 3).

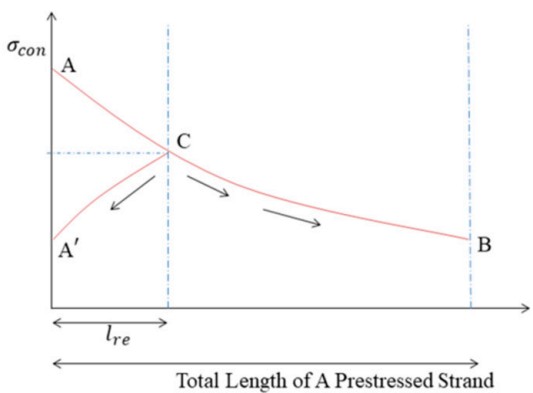

**Figure 3.** Distribution of strand stress caused by retraction.

The circumferential micro-segment of the prestressed steel strand is regarded as the research object, where the corresponding angle is $d\theta$ (Figure 4). Assuming that the normal stress of the steel

strand in the micro-segment is evenly distributed, a differential equation can be established according to the static equilibrium conditions:

$$T \cdot sin\left(\frac{d\theta}{2}\right) + (T + dT) \cdot sin\left(\frac{d\theta}{2}\right) - dP = 0 \tag{1}$$

where $T$ and $P$ stand for the tension force and the normal pressure of the strand, respectively.

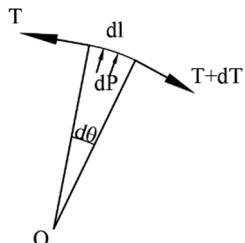

**Figure 4.** Stress of a micro-segment of a strand.

Higher variables were omitted, taking $\frac{d\theta}{2} = sin\left(\frac{d\theta}{2}\right)$. Equation (1) can be simplified to $Td\theta = dP$. The equation describing the momentary balance for rotation around the center of curvature, $O$, can be written as follows:

$$r_{st} \cdot \mu dP + r_{st} dT = 0 \tag{2}$$

where $T = T_0$ when $\theta = 0$, thus, $T = T_0 e^{-\mu\theta}$. $r_{st}$, where $r_{st}$ is the calculated radius of the strand wrapped outside the pipe (m) and $\mu$ is the friction coefficient between the prestressed strands and the outer surface of the deteriorating pipe. The influencing factors of $\mu$ mainly include the type of steel, the type of lubricating grease, the materials wrapped outside, and the quality control of the construction. Here, $\mu$ ranges from 0.08 to 0.12, with a mean value of 0.1.

The stress for an arbitrary cross section is calculated as per Equation (3):

$$\sigma = \sigma_{st} e^{-\mu\theta} \tag{3}$$

where $\sigma_{st}$ is the tension stress of prestressed steel strands (N/mm$^2$). $\sigma_{st} = f_{st,t} \cdot \alpha$, in which $\alpha$ is the control coefficient for the tension of the steel strands (N/mm$^2$). Normally, this value ranges between 0 and 0.75 [17–19]. $f_{st,t}$ is the nominal tensile strength of the prestressed strand (N/mm$^2$).

The stress at the end section of the retraction length can be written as follows:

$$\sigma_{re} = \sigma_{st} e^{-\mu\theta_{re}}. \tag{4}$$

The length reduction of the strand caused by the anchor deformation and the clip retraction, $\Delta l_{re}$, can be expressed by Equation (5):

$$\Delta l_{re} = \int_0^{\theta_{re}} \frac{\sigma_{l2} r_{st}}{E_{st}} d\theta = \frac{2 r_{st} \sigma_{st} \left[1 - e^{-\mu\theta_{re}}(1 + \mu\theta_{re})\right]}{\mu E_{st}} \tag{5}$$

where $E_{st}$ is the elastic modulus of the adopted steel strand (N/mm$^2$) and $e^{-\mu\theta_{re}}$ is expanded into a power series according to the Taylor formula. Only the first three terms of the formula have sufficient precision, since $\mu\theta_{re}$ is adequately small, which is given by Equation (6):

$$e^{-\mu\theta_{re}} = 1 - \mu\theta_{re} + \frac{(\mu\theta_{re})^2}{2}. \tag{6}$$

Equation (7) can be derived by incorporating Equation (6) into Equation (5) and omitting the high micro $(\mu\theta_{re})^3$:

$$\Delta l_{re} = \frac{\mu r_{st}\sigma_{st}\theta_{re}^2}{E_{st}} \tag{7}$$

The correspondence between the retraction length, $l_{re}$, and the retraction angle, $\theta_{re}$, is represented as follows:

$$l_{re} = r_{st}\theta_{re}. \tag{8}$$

Therefore, the retraction length, $l_{re}$, and its corresponding angle, $\theta_{re}$, can be given by Equations (8) and (9).

$$l_{re} = \sqrt{\frac{\Delta l_{re}E_{st}r_{st}}{\mu\sigma_{st}}} \tag{9}$$

The various types of anchorage used with steel strands were classified as plug and cone, straight sleeve, contoured sleeve, metal overlay, and split wedge anchorages. The value of $\Delta l_{re}$ varies with the type of anchor.

### 2.1.2. Prestress Loss Caused by Friction Resistance, $\sigma_{l1}$

The prestress loss caused by the friction resistance, $\sigma_{l1}$, can be calculated based on the consideration of two parts, namely, the bending loss and the deviation loss. The radial pressing force, $\sigma_r$, is produced between the strand and the pipe wall by prestressed strands, thereby resulting in extrusion friction. The bending loss accounted for a large proportion of the total friction loss.

Based on the assumption of a rigid body, we hypothesized that the pressure between the strand and the pipe wall would be uniformly distributed [20], and that elastic deformation would occur when the two elastic bodies were pressed into contact with each other. The stress between the contact surfaces is ellipsoidal, and its value can be related to the radius of curvature and the elastic modulus of the contact object. It is not accurate enough to consider the contact stress as uniformly distributed under normal contact pressure due to the large tensile force of the prestressed steel strands.

The scope of the bending loss can be related to the retraction length, $l_{re}$. We can assume that the normal stress between the strands and the pipe wall would be distributed as a trigonometric function [21], as illustrated in Figure 5 and Equation (10).

$$p_{(\alpha)} = p_0 \cos^2\left(\frac{\pi}{\theta}\alpha\right) \tag{10}$$

where $\cos^2\left(\frac{\pi}{\theta}\alpha\right) = \begin{cases} 0 \ \alpha = \frac{\theta}{2} \\ 1 \ \alpha = 0 \\ 0 \ \alpha = -\frac{\theta}{2} \end{cases}$

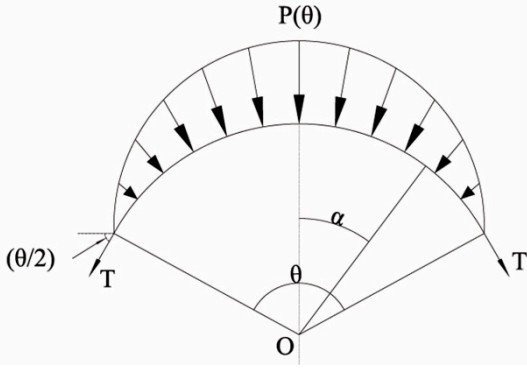

**Figure 5.** Distribution of the normal stress of the pipe wall excluding the friction.

A balance of forces in the z-direction can be established by Equation (11). From Equation (11), we derived Equation (12). Therefore, the normal stress can be calculated using Equation (13):

$$T \cdot \sin\left(\frac{\theta}{2}\right) + (T + dT) \cdot \sin\left(\frac{\theta}{2}\right) + 2 \int_0^{\frac{\theta}{2}} p_{(\alpha)} \cdot \cos \alpha \, dl = 0 \tag{11}$$

$$p_0 = \frac{\sin\frac{\theta}{2}}{\int_0^{\frac{\theta}{2}} \cos^2\left(\frac{\pi}{\theta}\alpha\right) \cos \alpha \, d\alpha} \times \frac{T}{R} \tag{12}$$

where $\int_0^{\frac{\theta}{2}} \cos^2\left(\frac{\pi}{\theta}\alpha\right) \cos \alpha \, d\alpha = \left(\cos^2\left(\frac{\pi}{\theta}\alpha\right) \sin \alpha\right)\Big|_0^{\frac{\theta}{2}} + \frac{\pi}{\theta} \int_0^{\frac{\theta}{2}} \sin \alpha \sin\left(\frac{2\pi}{\theta}\alpha\right) d\alpha = \frac{2\left(\frac{\pi}{\theta}\right)^2 \sin\frac{\theta}{2}}{\left(\frac{2\pi}{\theta}\right)^2 - 1}$.

$$p_{(\alpha)} = \left(2 - \frac{\theta^2}{2\pi^2}\right) \cdot \cos^2\left(\frac{\pi}{\theta}\alpha\right) \cdot \frac{\sigma_0}{R} \tag{13}$$

The prestress loss related to the bending loss, $F$, during the tensioning operation is depicted in Equation (14).

$$F = 2 \int_0^{\frac{\theta}{2}} \mu p_{(\theta)} \cdot dl = \mu \theta \sigma_0 \left(1 - \frac{\theta^2}{4\pi^2}\right) \tag{14}$$

The deviation loss stems from errors in pipe positioning and installation, which causes friction between the force rib and the pipe material, thereby forming contact friction. The deviation loss occupies a small proportion of the total friction loss. The correction coefficient, $c_1$, is involved here, and the deviation loss is not separately calculated in this paper. As a result, the total prestress loss caused by the friction resistance can be calculated, as displayed in Equation (15).

$$\sigma_{l1} = c_1 F \tag{15}$$

where $c_1$ is the correction coefficient, accounting for the bending loss and the deviation loss, and is usually in the range of 1 to 1.3 [19].

### 2.1.3. Prestress Loss Caused by Anchorage Deformation, $\sigma_{l2}$

The prestress loss caused by deformation at the end of the anchorage should be taken into consideration. This refers to the prestress loss caused by the deformation of the anchor and the retraction of the clip due to the concentrated stress. A slip at the anchorage depends on the particular prestressing system adopted and is not a function of time. This loss can be written as per Equation (16):

$$\sigma_{l2} = E_{st} \frac{\Delta l_{re}}{2\pi r_{st}}. \tag{16}$$

### 2.1.4. Prestress Loss Caused by the Elastic Compression of Concrete During Batch Tensioning, $\sigma_{l3}$

The prestress loss can be adjusted using certain construction technologies, including ultra-tensioning and repeated tensioning. For example, the tensioning can be started from the middle of the pipeline and gradually pulled symmetrically to both sides when batch tensioning is adopted. The steel strands that would later be tensioned would cause elastic compression deformation of the concrete, which would contribute to the prestress loss of the previously anchored strands. This prestress loss can be simplified by the following formula [21]:

$$\sigma_{l3} = \frac{m-1}{2m} n_y \sigma_{h1} \tag{17}$$

where $m$ is the total number of batches and $\sigma_{h1}$ is the normal stress of the concrete produced by the combined force of the steel strands at the action point (the center of gravity of all steel strands), which

is equal to the sum of the normal stresses of the concrete produced by the batch of steel strands, which is $\Delta\sigma_{h1}$. That is, $\sigma_{h1} = \sum \Delta\sigma_{h1} = m\Delta\sigma_{h1}$. $n_y$ is the combined force of all of the steel strands. $\Delta\sigma_{h1}$ is the normal stress of the concrete generated by the subsequent batch of steel strands at the center of gravity of the first tensioned steel strand, as calculated by the following formula: $\Delta\sigma_{h1} = \frac{n}{m}\left(\frac{1}{A_j} + \frac{e_y \cdot y_i}{I_j}\right)$.

During engineering, ultra-tensioning or repeated tensioning technologies can be utilized in the first several batches of strands, so that the actual effective prestress of the pipe is substantially equal to the design level. After ultra-tensioning or repeated tensioning, the prestress loss caused by the elastic compression of the concrete during batch tensioning, $\sigma_{l3}$, is considered to be approximately zero.

### 2.1.5. Prestress Loss Due to Crack Reduction and Closure, $\sigma_{l4}$

In our experiment, due to the restraining effect of the prestressed steel strand, the cracks of the concrete core were reduced to some extent, or even closed. The reduction of the circumference of the pipe led to the prestress loss of the strands. Therefore, the change in the maximum width of the visible cracks in the concrete core can be utilized to estimate the prestress loss caused by crack reduction and closure.

The change in the maximum width of the cracks corresponds to the change in the length of the prestressed steel strands (Figure 6). The prestressed steel strand is in the elastic phase, and the stress is proportional to its strain. Therefore, the prestress loss, $\sigma_{l4}$, caused by the crack reduction and closure, can calculated according to the following equation.

$$l_i = \frac{r_{st}}{D/2}w_i \tag{18}$$

$$\frac{\sigma_0}{\sigma_0 - \sigma_{l4}} = \frac{2\pi r_{st} + l_1}{2\pi r_{st} + l_2} \tag{19}$$

where $w_i$ is the maximum width of the visible cracks in the concrete core (m), $i = 1, 2$ for the condition before and after the reinforcement, $l_i$ is the length of prestressed strands corresponding to the maximum width of the cracks in the concrete core before and after the reinforcement (m), and $D$ is the outer diameter of the concrete core (*m*).

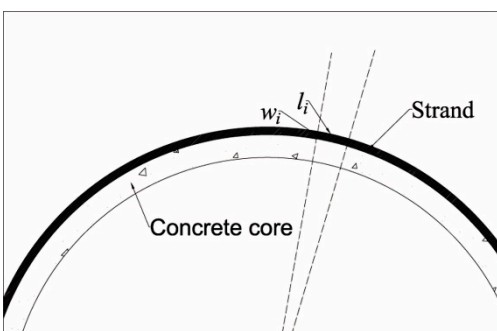

**Figure 6.** Correspondence between crack width and strand length.

Therefore, the prestress loss due to the crack reduction and closure, $\sigma_{l4}$, can be defined as Equation (20):

$$\sigma_{l4} = \frac{\sigma_0(w_1 - w_2)}{\pi D + w_1} \tag{20}$$

### 2.1.6. Prestress Loss Caused by Shrinkage and Creep of Concrete, $\sigma_{l5}$

The shrinkage and creep of the original concrete pipe were involved in the calculation of the prestressed stress of the prestressing wires. The shrinkage and creep of the reinforced pipe have been

basically completed before the reinforcement, and, as such, no further repeated calculations were performed for the prestress loss caused by shrinkage and creep of concrete, i.e., $\sigma_{l5} = 0$.

### 2.1.7. Prestress Loss Due to Long-Term Relaxation of the Strand, $\sigma_{l6}$

The deformation of the strand will change with time, and the stress will decrease accordingly when the strand is subjected to a constant external force, which is the prestress loss due to the long-term relaxation of the strand. The greater the tensile force of the steel strand, the more obvious the stress relaxation effect is. The relaxation generally occurred earlier in the process, without considering the quality of the strands. This effect can be basically completed after one year, and then gradually calmed. The relaxation loss, $\sigma_{l6}$, is related to the relaxation coefficient, $k$, and can calculated according to the following formula:

$$\sigma_{l6} = k\sigma_{st} \tag{21}$$

where $k$ is the relaxation coefficient and is related to the quality of the steel. For the cold-drawn thick steel bar, $k$ is taken as 0.05 for one-time tensioning and 0.035 for ultra-tensioning. As for the steel wires and steel strands, $k$ is considered to be 0.07 for one-time tensioning and 0.045 for ultra-tensioning [17]. For low-relaxation steel wires, the value of $k$ can be taken to be 0.002 when no data are available, which we have learned from our experience.

The total prestress loss of the prestressed wires can calculate by Equation (22):

$$\sigma_{st,l} = \sum_{1}^{6} \sigma_i \tag{22}$$

### 2.2. Calculation of Area of Prestressed Steel Strands

According to the study by Zarghamee M. [22–25], the cracking of PCCPs under combined loads mainly occurs at (1) the bottom of the inner surface of the concrete core, (2) the top of the inner surface of the concrete core, and (3) the spring-line of the outer surface of the concrete core. Therefore, these three sections are defined as dangerous sections. The area of prestressed steel strands can be determined under the assumption of a complete loss of prestress of prestressing wires.

### 2.2.1. Stress of PCCPs under Combined Loads

The combined loads acting on the pipe include the vertical earth pressure at the top of the pipe, $F_{sv,k}$, the lateral earth pressure, $F_{ep,k}$, the ground pile load, the weight of the pipe, $G_{1k}$, the weight of fluid in the pipe, $G_{wk}$, and the variable load.

The values of $F_{sv,k}$ and $F_{ep,k}$ are calculated according to Marston's theory [17] and Rankine's earth pressure theory [26], respectively. The variable load can be regarded as the ground stacking load, and its standard value is defined as $q_{mk} = 10 \text{ kN/m}$.

The weight of the pipe can be written as per Equation (23):

$$G_{1k} = \pi r_G (D_i + h_c) h_c \tag{23}$$

where $D_i$ is the inner diameter of the pipe (m), $h_c$ is the thickness of concrete core (m), and $r_G$ is the gravity density of the pipe (kN/m$^3$).

The weight of fluid in the pipe, $G_{wk}$, can be calculated by Equation (24):

$$G_{wk} = \frac{r_W \pi D_i^2}{4} \tag{24}$$

where $r_w$ is the gravity density of the fluid in the pipe (kN/m$^3$).

2.2.2. Calculation of the Area of Prestressed Strands, Considering the Concrete Core Compression of Ultimate Limit States

According to Chinese specifications [27,28], the design requirements for the calculation of ultimate limit states under the external soil load, weight of pipe, weight of fluid, and other variable loads are detailed. The design value of the maximum bending moment of the pipe at the spring-line, $M_{max}^l$, can be calculated using Equation (25). The value of $M_{max}^l$ is negative, indicating that the outer surface of the concrete core is subjected to tension. The absolute value can be taken when the formula is substituted for reinforcement. The design value of the maximum axial tension of the pipe at the spring-line, $N^l$, is written as per Equation (26):

$$M_{max}^l = \gamma_0 r \left[ k_{vm} \left( \gamma_{G3} F_{sv,k} + \psi_c \gamma_{Q2} q_{vk} D_1 \right) + k_{hm} \gamma_{G4} F_{ep,k} D_1 + k_{wm} \gamma_{G2} G_{wk} + k_{gm} \gamma_{G1} G_{1k} \right] \tag{25}$$

$$N^l = \gamma_0 \left[ \psi_c \gamma_{Q1} P_d r \times 10^{-3} - 0.5 \left( F_{sv,k} + \psi_c q_{vk} D_1 \right) \right] \tag{26}$$

where $\gamma_0$ is the factor of importance. It varies with the structure and the layout of the pipes. The value of $\gamma_0$ is generally 1.1. For side-by-side pipelines, the $\gamma_0$ value should be taken as 1.0, in particular. Moreover, the value of $\gamma_0$ should also be taken as 1.0 for pipes with storage facilities or those which are used for drainage. $r$ is the calculated radius of the pipe (m), and $k_{vm}$, $k_{hm}$, $k_{wm}$, $k_{gm}$ represent the bending moment coefficient of the bending moment at the spring-line of the pipe under the vertical earth pressure, lateral earth pressure, the weight of fluid inside the pipe, and the weight of the pipe, respectively. These factors were determined according to Appendix E [27]. The $k_{gm}$ of the arc-shaped soil bedding can be adopted according to the data of the bedding angle of 20°. $\gamma_{Gi}$ and $\gamma_{Qj}$ are the partial coefficients under the permanent load i and the variable load j. $\psi_c$ is the combination coefficient of the variable loads and usually takes the value of 0.9. $D_1$ is the outer diameter of the pipe (m). $P_d$ is the designed water pressure (N/mm²).

The area of the prestressed strands of ultimate limit states should be calculated by Equation (27):

$$A_{st} \geq \frac{\lambda_y}{f_{pyk}} \left( N^l + \frac{M_{max}^l}{d_0} - A_{sc} f'_{yy} \right) \tag{27}$$

where $\lambda_y$ is the comprehensive adjustment factor of the PCCP, $f_{pyk}$ is the design strength of prestressed strands (N/mm²), $d_0$ is the distance from the prestressed strand to the center of gravity of the pipe(m), $A_{sc}$ is the area of the cylinder per unit length (m²/m), and $f'_{yy}$ is the design strength of the cylinder (N/mm²).

2.2.3. Checking Calculation of Prestressed Strands Considering the Concrete Core Compression of Serviceability Limit States

The maximum bending moment of the pipe at the top or the bottom, $M_{pms}$, is calculated as per Equation (28). The value of $M_{pms}$ is negative, indicating that the outer surface of the concrete core is subjected to tension. The absolute value is taken when substituting the following equations. The axial tension of the pipe wall, $N_{ps}$, is written as per Equation (29):

$$M_{pms} = \gamma_0 \left[ k_{vm} \left( F_{sv,k} + \psi_c q_{vk} D_1 \right) + k_{hm} F_{ep,k} D_1 + k_{wm} G_{wk} + k_{gm} G_{1k} \right] \tag{28}$$

$$N_{ps} = \psi_c P_d r \times 10^{-3} \tag{29}$$

where $k_{vm}$, $k_{hm}$, $k_{wm}$, and $k_{gm}$ represent the bending moment coefficient of the bending moment at the top or the bottom of the pipe under the vertical earth pressure, lateral earth pressure, the weight of fluid inside the pipe, and the weight of the pipe, respectively. These factors can be determined according to Appendix E [27]. The $k_{gm}$ of the arc-shaped soil bedding can adopted, according to the data of the bedding angle of 20°.

The maximum tensile stress at the edge of the pipe at the bottom, $\sigma_{ss}$, is calculated as per Equation (30).

$$\sigma_{ss} = \frac{N_{ps}}{A_n} + \frac{M_{pms}}{\omega_c W_p} \tag{30}$$

where $A_n$ is the conversion area of the pipe section (including the cylinder, steel strands, and the mortar coating) (m$^2$/m). $\omega_c$ is the conversion coefficient of the elastic resistance moment of the tensioned edge of the pip -wall. $W_p$ is the momentary elastic resistance of the unconverted tension edge of the rectangular section of the pipe wall (m$^2$/m).

The effective prestress of the prestressed steel strands after the prestress loss, $\sigma'_{st}$, can be written as $\sigma'_{st} = \sigma_{st} - \sigma_{st,l}$.

Therefore, the area of prestressed strands of serviceability limit states should be calculated by Equation (31):

$$A_{st} \geq \left(\sigma_{ss} - K\gamma f_{ty}\right)\frac{A_n}{\sigma'_{st}} \tag{31}$$

where $K$ is the influence coefficient of concrete in the tension area, $\gamma$ is the plastic influence coefficient of concrete in the tension area, and $f_{ty}$ is the standard value of concrete tensile strength.

The area of prestressed strands needs to simultaneously meet the requirements outlined in Equations (27) and (31).

### 2.2.4. Checking Calculation of the Mortar Coating under Serviceability Limit States

Checking the calculation of mortar at the spring-line of the pipe should be carried out under serviceability limit states.

The maximum bending moment of the pipe at the spring-line, $M^l_{pms}$, can be calculated by Equation (32). The value of $M^l_{pms}$, is negative, indicating that the mortar coating is subjected to tension. The absolute value is taken when the following equations are substituted. The axial tension of the pipe at the spring-line, $N^l_{ps}$, can be written as per Equation (33):

$$M^l_{pms} = r\left[k_{vm}\left(F_{sv,k} + \psi_c q_{vk}D_1\right) + k_{hm}F_{ep,k}D_1 + k_{vm}G_{wk} + k_{gm}G_{1k}\right] \tag{32}$$

$$N^l_{ps} = \psi_c P_d r \times 10^3 - 0.5\left(F_{sv,k} + \psi_c q_{vk}D_1\right) \tag{33}$$

The maximum tensile stress at the edge of the pipe at the spring-line, $\sigma^l_{ss}$, can be calculated as per Equation (34):

$$\sigma^l_{ss} = \frac{N^l_{ps}}{A_n} + \frac{M^l_{pms}}{\omega_m W_p} \tag{34}$$

The maximum tensile stress at the edge of the mortar coating at the spring-line, $\sigma^l_{ss}$, should be less than its tensile strength (Equation (35)) under serviceability limit states. If not, Sections 2.2.2 and 2.2.3 should be repeated.

$$\sigma^l_{ss} \leq \alpha_m \varepsilon_{mt} E_m \tag{35}$$

where $\alpha_m$ is the design parameter of the mortar coating strain, which is equal to 5. $\varepsilon_{mt}$ is the strain of mortar coating when the strength reaches the tensile strength, and can be given as $\varepsilon_{mt} = \frac{f_{mt,k}}{E_m} \geq \frac{0.52\sqrt{f_{mc,k}}}{E_m}$.

### 2.2.5. Checking Calculation of Mortar Coating under Quasi-Permanent Limit States

Checking the calculation of mortar at the spring-line of the pipe should be carried out under quasi-permanent limit states.

The maximum bending moment of the pipe at the spring-line, $M^l_{pml}$, can be calculated as per Equation (36). The value of $M^l_{pml}$ is negative, indicating that the mortar coating is subjected to tension.

The absolute value is taken when substituting the following equations. The axial tension of the pipe at the spring-line, $N_{ps}^l$, is written as per Equation (37):

$$M_{pml}^l = r\left[k_{vm}\left(F_{sv,k} + \psi_{qv}q_{vk}D_1\right) + k_{hm}F_{ep,k}D_1 + k_{vm}G_{wk} + k_{gm}G_{1k}\right] \tag{36}$$

$$N_{pl}^l = \psi_{qw}P_d r \times 10^3 - 0.5\left(F_{sv,k} + \psi_{qv}q_{vk}D_1\right) \tag{37}$$

where $\psi_{qv}$, $\psi_{qw}$ is the quasi-permanent coefficient of vertical pressure generated by ground vehicle loads and the internal water pressure, respectively.

The maximum tensile stress at the edge of the pipe at the spring-line, $\sigma_{ls}^l$, is calculated as per Equation (38):

$$\sigma_{ls}^l = \frac{N_{pl}^l}{A_n} + \frac{M_{pml}^l}{\omega_m W_p}. \tag{38}$$

The maximum tensile stress at the edge of the mortar coating at the spring-line, $\sigma_{ss}^l$, should be less than its tensile strength (Equation (39)) under quasi-permanent limit states. If not, we return to Equations (2) and (3).

$$\sigma_{ls}^l \le \alpha_m' \varepsilon_{mt} E_m \tag{39}$$

where $\alpha_m'$ is the design parameter of strain for mortar coating and is equal to 4.

Above all, the area of prestressed strands per unit length, $A_{st}$, should be determined.

The prestressed steel strands are spirally wound at equal intervals. Thus, the center spacing of steel strands can be calculated by Equation (40):

$$l_{st} = A \times \frac{1000}{A_{st}} \tag{40}$$

where $A$ is the nominal section area, without polyethylene, of the adopted steel strand.

## 3. Applications

In order to verify the feasibility of the deduction, an example calculation of the theory and a prototype test were carried out on the same PCCP with broken wires. The specimen was an embedded prestressed concrete cylinder pipe (ECP) and the calculation process used is illustrated in Section 3.2. The center spacing of steel strands, calculated through the deduction, was then applied to the same pipe in a prototype test (Section 3.3).

### 3.1. Parameters of the Design and Materials

The theory and prototype tests were carried out on the same pipe. The geometric parameters of the adopted pipe are given as Table 1. Key parameters of the materials, involving the concrete, mortar and cylinder, are shown in Table 2.

**Table 1.** Geometric parameters of the embedded concrete pipe (ECP).

| Geometric Parameter | Value | Geometric Parameter | Value |
|---|---|---|---|
| Inner diameter of PCCP, $D_i$/mm | 2000 | Net thickness of mortar coating, $h_m$/mm | 25 |
| Thickness of core concrete, $h_c$/mm | 140 | Spacing between each wire, $l_s$/mm | 22.1 |
| Outer diameter of cylinder, $D_y$/mm | 2103 | Diameter of wires, $d_s$/mm | 6 |
| Thickness of cylinder, $t_y$/mm | 1.5 | Number of layers, $n$ | 1 |

**Table 2.** Key parameters of the materials.

| Key Parameter | Value | Key Parameter | Value |
|---|---|---|---|
| Designed 28-day compressive strength of the core concrete, $f_c'$/(N/mm$^2$) | 44 | Modulus of concrete, $E_c$/(N/mm$^2$) | $3.55 \times 10^4$ |
| Standard compressive strength of mortar, $f_{mc,k}$/(N/mm$^2$) | 45 | Modulus of mortar, $E_m$/(N/mm$^2$) | $2.416 \times 10^4$ |
| Poisson's ratio of concrete, $v_c$ | 0.167 | Modulus of cylinder, $E_y$/(N/mm$^2$) | $2.068 \times 10^5$ |
| Poisson's ratio of mortar, $v_m$ | 0.2 | Modulus of wire, $E_s$/(N/mm$^2$) | $1.93 \times 10^5$ |
| Minimum tensile strength of the prestressed wire, $f_{su}$/(N/mm$^2$) | 1570 | Designed tensile or compressive yield strength of steel cylinder, $f_{yy}$/(N/mm$^2$) | 227.5 |
| Gross wrapping tensile stress in wire, $f_{sg}$/(N/mm$^2$) | $0.75 f_{su}$ | Designed tensile strength of steel cylinder at pipe burst, $f_{yy}'$/(N/mm$^2$) | 215 |
| Design tensile strength of core concrete, $f_t'$/(N/mm$^2$) | 1.95 | Unit weight of the pipe, $\gamma_c$/kN/m$^3$ | 25 |
| Standard tensile strength of core concrete, $f_{ty}$/(N/mm$^2$) | 2.75 | Unit weight of mortar, $\gamma_m$/kN/m$^3$ | 23.5 |
| Unit weight of backfill soil, $\gamma_s$/kN/m$^3$ | 18 | Unit weight of water, $\gamma_w$/kN/m$^3$ | 10 |

As for the parameters of load, the internal working pressure used was $P_w = 0.6 \, \text{N/m}^2$. The internal transient pressure was $\Delta H_r = \max(0.4 P_w, 276 \, \text{kPa}) = 0.276 \, \text{N/mm}^2$. The internal design pressure was $P_d = P_w + \Delta H_r = 0.876 \, \text{MPa} \approx 0.9 \, \text{N/mm}^2$. The thickness of soil above the top of the pipe was $H = 3$ m. The bedding angle was 90°. The type of installation was trench-type with a positive projecting embankment. The standard value of the ground stacking load was $q_{mk} = 10 \text{kN/m}^2$.

The parameters of environment are shown as follows: The average relative humidity of the storage environment was 70% RH, the time in outdoor storage was $t_1 = 270$ d. Burial time after outdoor storage was $t_2 = 1080$ d.

Key parameters of the adopted strand are given in Table 3.

**Table 3.** Key parameters of the adopted strand.

| Key Parameter | Value | Key Parameter | Value |
|---|---|---|---|
| Nominal diameter without polyethylene d$_{st}$/mm | 15.2 | Control coefficient for the tensioning of the steel strands, $\alpha$ | 0.63 |
| Nominal section area without PE, $A$/mm$^2$ | 140 | Tension stress of the prestressed steel strands, $\sigma_{st}$/(N/mm$^2$) | 1171.8 |
| Nominal tensile strength, $f_{st,t}$/( N/mm$^2$) | 1860 | Standard tensile yield strength, $f_{pyk}$/(N/mm$^2$) | 1580 |
| Modulus of the strand, $E_{st}$/( N/mm$^2$) | $1.95 \times 10^5$ | Designed tensile yield strength, $f_{py}$/(N/mm$^2$) | 1110 [17] |

### 3.2. Example Calculation

#### 3.2.1. Calculation of Prestress Loss, $\sigma_{st,l}$

For the utilized split wedge without jacking force, $\Delta l_{re} = 6$ mm (measured in the prototype test [29]). Given $\mu = 0.1$, the calculated radius of the strand wrapped outside the pipe, $r_{st}$, and the retraction length, $l_{re}$, can be known as follows:

$$r_{st} = \frac{D_i}{2} + h_c + \frac{d_{st}}{2} = 1.1726 \, \text{m}, l_{re} = \sqrt{\frac{\Delta l_{re} E_{st} r_{st}}{\mu \sigma_{st}}} = 3.4217 \, \text{m}$$

The corresponding angle of the retraction length, $\theta_{re}$, is $\pi$, which is consistent with the value of $\theta$, indicating that the assumption is reasonable (Equation (10)).

The prestress loss caused by friction resistance, anchorage deformation, elastic compression of concrete during batch tensioning, crack reduction and closure, shrinkage and creep of concrete, and long-term relaxation of the strand is given in Table 4.

**Table 4.** The calculation results of the prestress loss.

| Item | Value |
|---|---|
| Prestress loss related to the bending loss, $F$ | 276.099 |
| Correction coefficient accounting for the bending loss and the deviation loss, $c_1$ | 1.01 |
| Prestress loss caused by friction resistance, $\sigma_{l1}/(\,N/mm^2)$ | 278.860 |
| Prestress loss caused by anchorage deformation, $\sigma_{l2}/(\,N/mm^2)$ | 158.802 |
| Prestress loss caused by elastic compression of concrete during batch tensioning, $\sigma_{l3}/(\,N/mm^2)$ | 0 |
| Maximum width of the visible cracks in the concrete core before the reinforcement, $w_1/m$ | 0.0022 |
| Maximum width of the visible cracks in the concrete core after the reinforcement, $w_2/m$ | 0.0001 |
| Prestress loss due to the crack reduction and closure, $\sigma_{l4}/(\,N/mm^2)$ | 0.3434 |
| Prestress loss caused by shrinkage and creep of concrete, $\sigma_{l5}/(\,N/mm^2)$ | 0 |
| The relaxation coefficient of the strand, k | 0.045 |
| Prestress loss due to long-term relaxation of the strand, $\sigma_{l6}/(\,N/mm^2)$ | 52.731 |
| Total prestress loss of prestressed wires, $\sigma_{st,l}/(\,N/mm^2)$ | 490.74 |

### 3.2.2. Stress of PCCP under Combined Loads

The stress of the adopted PCCP under combined loads, involving the vertical earth pressure at the top of the pipe, the lateral earth pressure, the variable load, weight of the pipe, and weight of water in the pipe, is presented in Table 5.

**Table 5.** The calculation results of the stress.

| Item | Value |
|---|---|
| Vertical earth pressure at the top of the pipe, $F_{sv,k}/(kN/m)$ | 164.245 |
| Lateral earth pressure, $F_{ep,k}/(kN/m)$ | 25.026 |
| Variable load, $q_{mk}/(kN/m)$ | 10 |
| Weight of the pipe, $G_{1k}/(kN/m)$ | 29.157 |
| Weight of water in the pipe/$(kN/m)$ | 31.416 |

### 3.2.3. Calculation of Area of Prestressed Strands, Considering the Concrete Core Compression of Ultimate Limit States

Assuming that the area of prestressed strands is $A_{st} = 2223$ mm$^2$/m, the calculation process of the area of prestressed strands, considering the concrete core compression of ultimate limit states, can be depicted in Table 6.

The value of $M_{max}^l$ is negative, indicating that the outer surface of the concrete core is subjected to tension. The absolute value is taken when substituting the formula for reinforcement.

Therefore, $A_{st} \geq \frac{\lambda_y}{f_{pyk}}\left(N^l + \frac{M_{max}^l}{d_0} - A_{sc}f'_{yy}\right) = 1069.413$ m$^2$/m

**Table 6.** The calculation results of the stress.

| Item | Value |
|---|---|
| Thickness of the pipe, $T$/m | 0.171 |
| Calculated radius of the pipe, $r$/m | 1.0855 |
| Outer diameter of the pipe, $D_1$/m | 2.342 |
| Combination coefficient of the variable loads, $\psi_c$ | 0.9 |
| The factor of importance for two side-by-side pipelines, $\gamma_0$ | 1 |
| Design value of the maximum bending moment of the pipe at the spring-line, $M^l_{max}$/(kN·m/m) | −33.998 |
| Design value of the maximum axial tension of the pipe at the spring-line, $N^l$/(kN/m) | 1111.712 |
| Width of calculated section, $B$/m | 1 |
| Ratio of modulus of the strand to the concrete, $n_{st}$ | 5.83 |
| Ratio of modulus of the cylinder to the concrete, $n_y$ | 5.49 |
| Ratio of modulus of the mortar to the concrete, $n_m$ | 0.68 |
| Conversion area of pipe section (including cylinder, steel strands and the mortar coating), $A_n$/m$^2$ | 0.1792 |
| Cross sectional area moment of the cross section of the concrete core, mortar, steel cylinder and prestressed steel strand on the inner surface of the pipe wall, $S_n$/m$^3$ | 0.01496 |
| Cross sectional area of the cylinder for unit pipe length, $A_{sc}$/(m$^2$/m) | 0.0015 |
| Distance from the mandrel to the inner surface of the pipe wall after the conversion, $y_0$ | 0.08347 |
| Distance from the center of the prestressed steel strand to the center of gravity of the pipe wall section, $d_0$/m | 0.06412 |
| Comprehensive adjustment factor for ECP whose diameter is larger than 1600 mm, $\lambda_y$ | 0.9 |

Notes: $T = h_c + h_m + d_s$, $r = \frac{D_i + T}{2}$, $D_1 = D_i + 2T$, $A_n = Bh_c + \left(n_y - B\right)Bt_y + (n_{st} - n_m)A_{st} + n_m B(T - t)$, $S_n = \frac{Bh_c^2}{2} + \left(n_y - \right)Bt_y\frac{(D_y - D_i - t_y)}{2} + (n_{st} - n_m) * A_{st} + n_m B(T - h_c)\left(\frac{T - h_c}{2} + h_c\right)$, $A_{sc} = Bt_y$, $y_0 = \frac{A_n}{S_n}$, $d_0 = h_c + \frac{d_{st}}{2} \times 10^{-0} - y_0$.

### 3.2.4. Checking Calculation of Prestressed Strands Considering the Concrete Core Compression of Serviceability Limit States

The conversion coefficient of the elastic resistance moment of the tensioned edge of the pipe wall can be obtained by interpolation, where $\omega_c = 1.017$ and $\omega_m = 0.9932$. The checking calculation process of the prestressed strands, considering the concrete core compression of serviceability limit states, is depicted in Table 7.

**Table 7.** The calculation results of the stress.

| Item | Value |
|---|---|
| Maximum bending moment of the pipe at the bottom, $M_{pms}$/(kN·m/m) | 36.316 |
| Maximum bending moment of the pipe at the top, $M'_{pms}$/(kN·m/m) | 23.423 |
| Axial tension of the pipe wall, $N_{ps}$/(kN/m) | 879.255 |
| Elastic resistance moment of the unconverted tension edge of the rectangular section of the pipe wall, $W_p$/(m$^2$/m) | 0.00487 |
| Maximum tensile stress at the edge of the pipe at the bottom, $\sigma_{ss}$/(N/mm$^2$) | 12.107 |
| Effective prestress of the prestressed steel strands apart from the prestress loss, $\sigma'_{st}$/(N/mm$^2$) | 681.06 |
| Influence coefficient of concrete in tension area, $K$ | 1.2239 |
| Plastic influence coefficient of concrete in tension area, $\gamma$ | 1.75 |

Note: $K = 0.2449\frac{M_{pms}}{\omega_c W_p f_{ty}} + 0.5714$.

Therefore, the area of prestressed strands should meet the requirement of $A_{st} \geq \left(\sigma_{ss} - K\gamma f_{ty}\right)\frac{A_n}{\sigma'_{st}} = 2222.300$ mm$^2$/m.

Above all, the area of prestressed strands is $A_{st} = 2223$ mm$^2$/m.

### 3.2.5. Checking Calculation of Mortar Coating under Serviceability Limit States

The maximum bending moment of the pipe at the spring-line was $M^l_{pms} = -24.897$ kN·m/m. The value of $M^l_{pms}$, is negative, indicating that the mortar coating was subjected to tension. The absolute value was taken when substituting the following equations. The axial tension of the pipe at the spring-line was $N^l_{ps} = 769.388$ kN/m The maximum tensile stress at the edge of the pipe at the spring-line was $\sigma^l_{ss} = 9.44$ N/mm$^2$. The strain of mortar coating was $\varepsilon_{mt} = \frac{f_{mt,k}}{E_m} \geq \frac{0.52\sqrt{f_{mc,k}}}{E_m} = 0.0001444$. The design parameter of strain for the mortar coating was $\alpha_m = 5$. Thus, $\left(\alpha_m\varepsilon_{mt}E_m = 17.44 \text{ N/mm}^2\right) > \left(\sigma^l_{ss} = 9.44 \text{ N/mm}^2\right)$, indicating that the area of prestressed strands is able to meet the tensile requirement of the mortar coating under the serviceability limit states.

### 3.2.6. Checking Calculation of Mortar Coating under Quasi-Permanent Limit States

$\psi_{qv}$ and $\psi_{qw}$ are the quasi-permanent coefficient of vertical pressure generated by the ground vehicle loads and the internal water pressure, respectively. Here, $\psi_{qv} = 0.5$ and $\psi_{qw} = 0.72$. The maximum bending moment of the pipe at the spring-line was $M^l_{pml} = -23.422$ kN·m/m. The value of $M^l_{pml}$ is negative, indicating that the mortar coating was subjected to tension. The absolute value was taken when substituting the following equations. The axial tension of the pipe at the spring-line was $N^l_{pl} = 602.911$ kN/m. The maximum tensile stress at the edge of the pipe at the spring-line was $\sigma^l_{ls} = 8.21$ N/mm$^2$. The design parameter of strain for the mortar coating was $\alpha'_m = 4$. Therefore, $\left(\alpha'_m\varepsilon_{mt}E_m = 13.95 \text{ N/mm}^2\right) > \left(\sigma^l_{ss} = 8.21 \text{ N/mm}^2\right)$, indicating that the area of prestressed strands is able to meet the tensile requirement of the mortar coating under quasi-permanent limit states.

Above all, this is reasonable of the calculation result of the area of prestressed strands, which is $A_{st} = 2223$ mm$^2$/m. The center spacing of steel strands was $l_{st} = A \times \frac{1000}{A_{st}} = 62.99$ mm.

### 3.3. A Prototype Test

A prototype test of ECP reinforced by steel strands with the fixed spacing calculated in Section 3.2 was performed in an assembled apparatus (Figure 7). The apparatus was mainly constituted by two ECPs, whose internal diameters were 2000 mm [29]. The adopted pipes were exactly the same as those given in Section 3.1. The entire test process involved five load stages, namely, (1) increasing the internal water pressure to the working pressure (0–0.6 MPa), (2) cutting the prestressing wires manually until the cracks propagated in the concrete core (0.6 MPa), (3) decreasing the internal water pressure to the artesian pressure (0.6–0.2 MPa), (4) performing the tensioning operation after wrapping the strands externally around the pipe (0.2 MPa), and (5) increasing the internal water pressure to the original level (0.2–0.6 MPa). In most of the actual pipe failures modes, most pipes failed at 4 or 8 o'clock, not at the invert, crown, or spring-lines [29]. The position of 8 o'clock was chosen in this test for convenience (Figure 8).

Post-tensioning was designed with the theory conducted in Section 3.2, indicating that the target tensile strength was equal to 1171.8 MPa and the center spacing of steel strands was taken as 62 mm. To prevent a prestress loss due to the retraction of clips and the stress relaxation of strands, excessive stretching is essential here. The tensioning process is divided into six stages, which were 20%, 25%, 50%, 75%, 100%, and 115%. Tensioning was performed simultaneously from both sides and in a symmetrical manner along the pipeline axis.

The statuses of each component of the pipe and the steel strands were measured by resistance strain gauges along the axial direction at inverted (360°), crown (180°), and spring-line (90°, 270°) orientations (Figure 8). Figure 9 exhibits the hoop strains in the concrete core before and after the reinforcement under the working pressure (0.6 MPa). The strains in the concrete core all showed a drastic drop after the process of tensioning. Moreover, the maximum width of the cracks in the outer concrete core at spring-line reduced from 2.2 mm to 0.1 mm after strengthening, as observed through field observation.

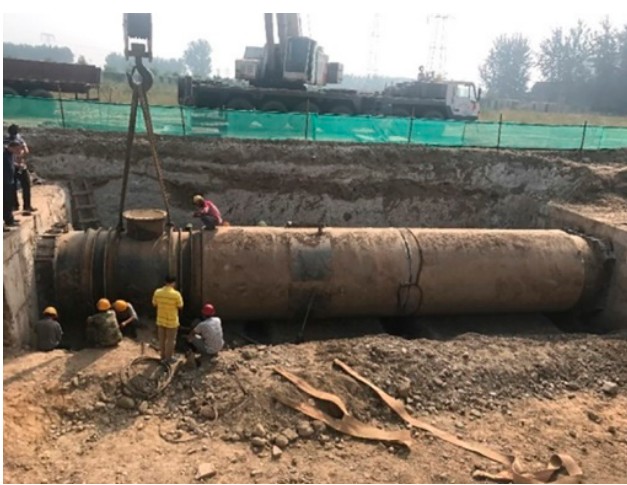

**Figure 7.** The spot photo of the test apparatus.

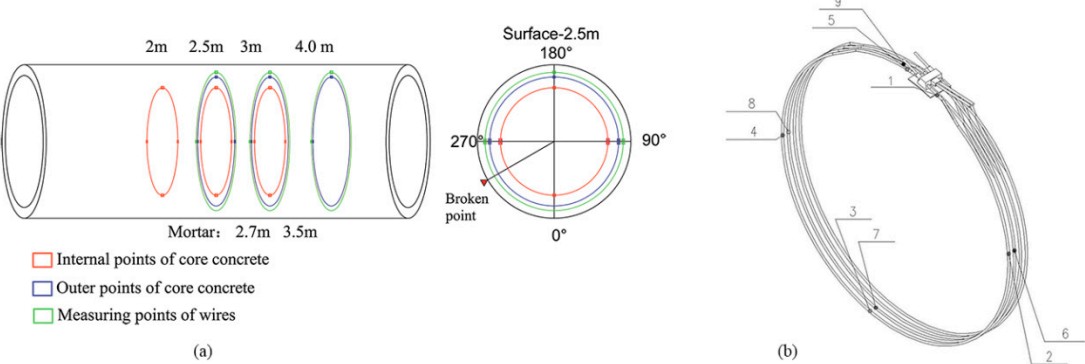

**Figure 8.** Layout of measuring points of (**a**) the pipe and (**b**) the steel strands.

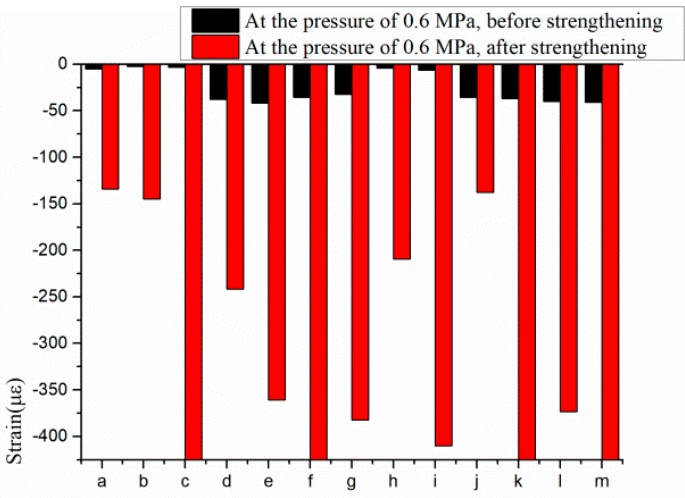

**Figure 9.** Comparison of strains in core concrete before and after the reinforcement: (**a**) 2.5 m inner concrete core at 0°; (**b**) 2.5 m inner concrete core at 90°; (**c**) 2.5 m inner concrete core at 180°; (**d**) 3 m inner concrete core at 0°; (**e**) 3 m inner concrete core at 90°; (**f**) 3 m inner concrete core at 180°; (**g**) 3 m inner concrete core at 270°; (**h**) 2.5 m outer concrete core at 0°; (**i**) 2.5 m outer concrete core at 90°; (**j**) 3 m outer concrete core at 90°; (**k**) 3 m outer concrete core at 180°; (**l**) 4 m outer concrete core at 90°; (**m**) 4 m outer concrete core at 180°.

The strengthened pipe was capable of sustaining the working pressure and the water tightness property was in a good state. The strains of the steel strands were all below the tensile strain level. The reinforcement of the PCCP with external prestressed steel strands was able to meet the strengthen requirement of the test. The rationality of the derivations in this paper were verified by the effective reinforcement effect with external prestressed steel strands.

## 4. Conclusions

A theoretical derivation was performed, aiming to determine the appropriate area of prestressed steel strands per unit length, and a prototype test was conducted to verify the rationality of the derivation in this study. The following conclusions can be drawn:

(1) The calculation formula for the prestress loss of different types of steel strands has been derived and the effective prestress of the prestressed steel strands can be determined.

(2) A stress calculation formula of the concrete core under the ultimate limit states and serviceability limit states was determined and used for calculation. The condition of the mortar coating under the serviceability limit and the quasi-permanent limit states was verified, and the reinforcement area of the steel strand was finally determined. This tentative derivation was applied to the reinforced pipe with broken wires (inner diameter of 2000 mm) to calculate the appropriate area of prestressed steel strands.

(3) The crack propagation in the concrete core was constrained by the strands and the test pipe was able to sustain the working pressure after strengthening. In addition, the maximum width of the cracks in the outer concrete core at the spring-line showed some closure because of the contribution of the strands. The bearing capacity of the prototype test was returned to the original design level and the behavior of the pipe was in accordance with the expectation of derivation.

**Author Contributions:** Conceptualization, T.D., S.X. and L.Z.; Methodology, T.D., L.Z., and B.C.; Software, T.D. and L.Z.; Writing—original draft preparation, L.Z.; Writing—review and editing, B.C., X.L., and M.L.; Funding acquisition, J.Y. and Q.Z.

**Funding:** This research was supported by the Beijing South-to-North Water Transfer Line Management Office (GXGLC-JSZX -2017-CG01), the National Natural Science Foundation of China (Grant No. 5097911), the Beijing Municipal Science and Technology Commission (Z141100006014058), and the China Institute of Water Resources and Hydropower Research (SM0145B632017).

**Acknowledgments:** The authors would like to thank the anonymous reviewers for their constructive suggestions to improve the quality of the paper.

**Conflicts of Interest:** The authors declare no conflict of interest.

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
