# Peer review of "Theoretical Study and Application of the Reinforcement of Prestressed Concrete Cylinder Pipes with External Prestressed Steel Strands"

_applsci, doi:10.3390/app9245532_

Round 1

Reviewer 1 Report

1. Minor grammatical errors. Review from a professional English speaking engineer of the same professional field is recommended.

2. For all the equations outlined in the paper, the explanations of the variables are written/explained in the text. This is not a correct format. The correct format would be;

Equation function    (number)

Where;

Variable 1 : definition

Variable 2 : definition

and so on. It is advised that the formatting for the mathematical equations (and there are quite a number of equations in the paper) be done correctly or rewritten in accordance to the MDPI; applied sciences format. 

3. Titles for Section 3 and Section 4 are both "Applications." This should be corrected, or the sections titles should be revised to be more specific. 

4. In contrast to the degree of details regarding the design calculations in the theoretical derivation (from Section 2.2 onwards), the results outlined in the experimental results (section 4?) is very lacking. Concluding that the design calculation (prediction) and test results for the PCCP prototype testing "agrees well" is not a scientific conclusion. Details of the experimental controls that led to this conclusion is completely absent in this paper. It is advised that section 4 be rewritten to include details of the test results to compare with the predicted design parameters (theoretical derivation). 

Author Response

Response to Reviewer 1 Comments

We would like to extend our sincere appreciation to the editors’ effort and reviewers’ constructive comments toward improving our manuscript. We have studied the reviewers’ comments carefully and have made extensive modifications to the previous manuscript which we hope to meet the requirements for publishing in Applied Sciences. The specific comments are addressed and explained in detail below and the revised manuscript according to the comments of the reviewers is attached.

Point 1: Minor grammatical errors. Review from a professional English speaking engineer of the same professional field is recommended.

Response 1: Thanks for your careful review and kind reminder. We feel really sorry for the grammatical errors. The manuscript has been modified by the official editing agency of MDPI. After that, we have studied the revised manuscript very carefully hoping that there is no grammatical error anymore. If there is any inappropriate, please do not hesitate to let us know. Thank you so much.

Point 2: For all the equations outlined in the paper, the explanations of the variables are written/explained in the text. This is not t correct format. The correct format would be;

Equation function    (number)

Where;

Variable 1 : definition

Variable 2 : definition

and so on. It is advised that the formatting for the mathematical equations (and there are quite a number of equations in the paper) be done correctly or rewritten in accordance to the MDPI; applied sciences format.

Response 2: Thanks for your careful review and kind reminder. We really appreciate your thoughtfulness. The format of variables has been modified carefully. The template provided by MDPI allows the variables to be written in the text or listed separately. So we have changed part of variables to be listed separately and simultaneously reserved some variables to be written in the text considering the length of the article. Thank you again for your advice.

Point 3: Titles for Section 3 and Section 4 are both "Applications." This should be corrected, or the sections titles should be revised to be more specific.

Response 3: Thanks for your careful review and kind reminder. The structure of the article has been rearranged in the revised manuscript. The Section 3 includes two parts(1) An example calculation; (2) A prototype test. An example calculation of the theory and a prototype test was carried out on the same PCCP with broken wires. The specimen was an embedded prestressed concrete cylinder pipe(ECP) and the parameters of the pipe were given in Section 3.1. The calculation process was illustrated in Section 3.2. The center spacing of steel strands calculated through the deduction was applied to the same pipe through a prototype test then(Section 3.3).

Point 4: In contrast to the degree of details regarding the design calculations in the theoretical derivation (from Section 2.2 onwards), the results outlined in the experimental results (section 4?) is very lacking. Concluding that the design calculation (prediction) and test results for the PCCP prototype testing "agrees well" is not a scientific conclusion. Details of the experimental controls that led to this conclusion is completely absent in this paper. It is advised that section 4 be rewritten to include details of the test results to compare with the predicted design parameters (theoretical derivation).

Response 4: Thanks for your careful review and kind reminder. We haven’t mentioned so many details of the prototype test considering that the details of the prototype test, including the test materials, test apparatus, test procedures and so on, have been published before[1]. We are really sorry for our thoughtless. The part of the prototype test in the original version was indeed too simple. In the revised version, we supplemented the content of the test, including the test procedures, operation details, monitoring arrangements and the test results.

[1]  Zhao L; Dou T; Cheng B; et al. Experimental Study on the Reinforcement of Prestressed Concrete Cylinder Pipes with External Prestressed Steel Strands. Applied Sciences, 2019,9,149.

Reviewer 2 Report

The authors have presented comprehensive work on the analysis of prestressed concrete cylinder pipes.

The derivations are quite extensive. I would suggest making the derivations more concise if possible but I do not have any recommendations on how to do so.

p3 line 2. Please clarify statement, "Since temperature is a post-tensioning method..."

p.4 second paragraph. Please clarify "Higher variables". Same with statement before equation 7 "omitting the high micro." How was the average friction coefficient determined?

Equations 25, 28, 32, and 36. When is the gamma importance factor needed or not needed?

p10. After Equation 40. Was PE predefined in the text?

Sections 3 and 4 have the same title. I think Section 3 should say something like 'Example Calculation.' If so, give introductory statement on the premise of the example.

Is Section 4 the physical description of the example in Section 3? If so, I would recommend combining Sections 3 and 4, putting the background and picture first, followed by calculation parameters and assumptions, then calculation results, and finally the verification statements.

I would recommend putting the results of the calculations in tabular form. There is not much text but rather calculation results.

Author Response

Response to Reviewer 2 Comments

We would like to extend our sincere appreciation to the editors’ effort and reviewers’ constructive comments toward improving our manuscript. We have studied the reviewers’ comments carefully and have made extensive modifications to the previous manuscript which we hope to meet the requirements for publishing in Applied Sciences. The specific comments are addressed and explained in detail below and the revised manuscript according to the comments of the reviewers is attached.

Point 1: The authors have presented comprehensive work on the analysis of prestressed concrete cylinder pipes. The derivations are quite extensive. I would suggest making the derivations more concise if possible but I do not have any recommendations on how to do so.

Response 1: Thanks for your careful review and kind reminder. The manuscript has been revised carefully several times. The process of the derivations has been modified and simplified in the revised manuscript. Thank you for your suggestions again.

Point 2: p3 line 2. Please clarify statement, "Since temperature is a post-tensioning method..."

Response 2: Thanks for your careful review and kind reminder. We are really sorry for the inappropriate language. This sentence has been modified in the revised version.

Since temperature is a post-tensioning method, the impact of this factor was removed from consideration in regard to reinforcement of PCCPs with external prestressed steel strands.

Since the reinforcement of PCCP with external prestressed steel strands is a post-tensioning method, the impact of temperature was removed from consideration in regard to the reinforcement of PCCPs with external prestressed steel strands.

Point 3: p.4 second paragraph. Please clarify "Higher variables". Same with statement before equation 7 "omitting the high micro."

Response 3: Thanks for your careful review and kind reminder. The details of this point can be seen in the file. 

Point 4: How was the average friction coefficient determined?

Response 4: Thanks for your careful review and kind reminder. We are really sorry for the inappropriate language. The influencing factors of u mainly included the type of steel, the type of lubricating grease, the materials wrapped outside, and the quality control of the construction. The value of u is independent of the length. The “average” has been deleted and the statement has been modified to be more precisely as follows.

u is the average friction coefficient between the prestressed strands and the outer surface of the deteriorating pipe. The influencing factors of u mainly included the type of steel, the type of lubricating grease, the materials wrapped outside, and the quality control of the construction.  ranged from 0.08 to 0.12 with a mean value of 0.1. The actual value of  generally changed with the length of the prestressed strand. In this paper, this variety was negligible and the average friction coefficient, , was adopted.

Point 5: Equations 25, 28, 32, and 36. When is the gamma importance factor needed or not needed?

Response 5: Thanks for your careful review and kind reminder.  is the factor of importance. We have not introduced this variable in detail considering the word limitation. Sorry for the ambiguity.  is a coefficient reflecting the importance of the pipeline structure. It varies with the structure and the layout of the pipes. The value of r is 1.1 generally. For side-by-side pipelines, the r  should be taken as 1.0 in particular. Moreover, the value of r should also be taken as 1.0 for pipes with storage facilities or used as drainage. The different values for different situations have been clarified in the revised version.

The added contents can be found in the file and the revised manuscript.

Point 6: p10. After Equation 40. Was PE predefined in the text?

Response 6: Thanks for your careful check and valuable suggestions. We feel really sorry for our unclearness. PE here stands for polyethylene in particular. The strand adopted is a non-bonded prestressed steel strand with double polyethylene cables(PE), as illustrated in Figure 1. The epoxy coating is applied to the base material of the strand at the fabrication stage, which provides a layer of protection against corrosion. There are double polyethylene cables outside the strand to ensure the strand′s corrosion resistance property. In addition, the double polyethylene cables are filled with anti-corrosive grease of ≥50 g/m to achieve free sliding between PE1 and PE2 with low friction. In general, each strand is protected by three anti-corrosion barriers. The abbreviation ‘PE’ has been modified to polyethylene in the revised manuscript.

Figure 1. The structural drawing of steel strands: 1—Steel wires; 2—Polyethylene cables; 3—Grease.

Point 7&8: Sections 3 and 4 have the same title. I think Section 3 should say something like 'Example Calculation.' If so, give introductory statement on the premise of the example. Is Section 4 the physical description of the example in Section 3? If so, I would recommend combining Sections 3 and 4, putting the background and picture first, followed by calculation parameters and assumptions, then calculation results, and finally the verification statements.

Response 7: Thanks for your careful review and valuable suggestions. We feel really sorry for our negligence. As you mentioned, Section 3 is ‘Example calculation’ of the deviation we conducted in Section 2 and Section 4 is a prototype test to verify the reinforcement effect with the center spacing of steel strands calculated by Section 3. The structure of the article has been rearranged in the revised manuscript. The Section 3 includes two parts(1) An example calculation; (2) A prototype test. An example calculation of the theory and a prototype test was carried out on the same PCCP with broken wires. The specimen was an embedded prestressed concrete cylinder pipe(ECP) and the parameters of the pipe were given in Section 3.1. The calculation process was illustrated in Section 3.2. The center spacing of steel strands calculated through the deduction was applied to the same pipe through a prototype test then(Section 3.3).

The introductory statement of Section 3 is shown as follows.

This theory deviated was applied to the reinforcement design of the PCCP with broken wires in order to verify the feasibility of this deviation. The specimen was an embedded prestressed concrete cylinder pipe(ECP) and the calculation process was illustrated.

Point 9: I would recommend putting the results of the calculations in tabular form. There is not much text but rather calculation results.

Response 9: Thanks for your careful review and kind reminder. The results of the calculations have been organized into tables according to your suggestions. Thank you again for your kind reminder.

Round 2

Reviewer 1 Report

The paper has been revised extensively and it is of the reviewer's opinion that the current version of the article is qualified for a publication.

This manuscript is a resubmission of an earlier submission. The following is a list of the peer review reports and author responses from that submission.